# Evolution of Early Postoperative Cardiac Rehabilitation in Patients with Acute Type A Aortic Dissection

**DOI:** 10.3390/jcm11082107

**Published:** 2022-04-09

**Authors:** Na Zhou, Gabriel Fortin, Maria Balice, Oksana Kovalska, Pascal Cristofini, Francois Ledru, Warner M. Mampuya, Marie-Christine Iliou

**Affiliations:** 1Department of Cardiac Rehabilitation and Secondary Prevention, Corentin Celton Hospital, 92130 Paris, France; tuaslapeche@hotmail.com (N.Z.); maria.balice@aphp.fr (M.B.); oksana.kovalska@aphp.fr (O.K.); pascal.cristofini@aphp.fr (P.C.); francois.ledru@aphp.fr (F.L.); 2Service de Cardiologie, Centre Hospitalier Universitaire de Sherbrooke, Sherbrooke, QC J1H 5H3, Canada; gabriel.fortin2@usherbrooke.ca (G.F.); warner.mampuya@usherbrooke.ca (W.M.M.)

**Keywords:** cardiac rehabilitation, exercise training, type A acute aortic dissection, blood pressure, shear stresses

## Abstract

Introduction: Surgically treated acute type A aortic dissection (ATAAD) patients are often restricted from physical exercise due to a lack of knowledge about safe blood pressure (BP) ranges. The aim of this study was to describe the evolution of early postoperative cardiac rehabilitation (CR) for patients with ATAAD. Methods: This is a retrospective study of 73 patients with ATAAD who were referred to the CR department after surgery. An incremental symptom-limited exercise stress test (ExT) on a cyclo-ergometer was performed before and after CR, which included continuous training and segmental muscle strengthening (five sessions/week). Systolic and diastolic blood pressure (SBP and DBP) were monitored before and after all exercise sessions. Results: The patients (78.1% male; 62.2 ± 12.7 years old; 54.8% hypertensive) started CR 26.2 ± 17.3 days after surgery. During 30.4 ±11.6 days, they underwent 14.5 ± 4.7 sessions of endurance cycling training, and 11.8 ± 4.3 sessions of segmental muscle strengthening. At the end of CR, the gain of workload during endurance training and functional capacity during ExT were 19.6 ± 10.2 watts and 1.2 ± 0.6 METs, respectively. The maximal BP reached during endurance training was 143 ± 14/88 ± 14 mmHg. The heart rate (HR) reserve improved from 20.2 ± 13.9 bpm to 33.2 ± 16.8 bpm while the resting HR decreased from 86.1 ± 17.4 bpm to 76.4 ± 13.3 bpm. Conclusion: Early post-operative exercise-based CR is feasible and safe in patients with surgically treated ATAAD. The CR effect is remarkable, but it requires a close BP monitoring and supervision by a cardiologist and physical therapist during training.

## 1. Introduction

Even if surgical techniques and perioperative care have significantly improved during the last decades, mortality for patients suffering from an acute type A aortic dissection (ATAAD) remains high, and is reported to be between 15% and 30% [1,2,3,4,5].

The 1- and 3-year survival rates in patients with successfully surgically treated ATAAD are estimated at around 96% ± 2% and 91% ± 4%, respectively [1]. Though enhanced acute management (i.e., medical therapies, open surgery, or endovascular approaches) has contributed to the improved survival, these patients remain at increased risk of aortopathy and associated cardiovascular events. These threats may be favorably or unfavorably modulated by regular exercise. Routine physical exercise performed at a safe level is important for all individuals, including patients who have undergone cardiac surgery [6]. However, surgically treated ATAAD patients are often restricted from physical exercise due to a lack of knowledge and fear associated with a potential rise of arterial blood pressure (BP) during exercise [7,8,9]. Moreover, in the early post-operative period, the aerobic capacity of patients is usually not compatible with current exercise recommendations [8].

It is unclear whether physical exercise increases the risk of progression of the patent false lumen, potentially causing aortic dissection expansion, aneurysm formation and rupture, or malperfusion syndrome. It is also unclear under which conditions physical exercise is advisable for such patients [10]. Nevertheless, it is well known that a sedentary lifestyle is a cause of endothelial dysfunction, and a risk factor for cardiovascular disease [11,12]. Exercise modifies blood flow, luminal shear stress, arterial pressure, and tangential wall stress, all of which can induce changes in arterial function, diameter, and wall thickness [6]. Notably, exercise training can increase retrograde wall shear stress, although its effect on endothelial cells is still uncertain. Ku et al. [10] demonstrated that retrograde wall shear stress is detrimental to endothelial function [11]. However, it is also plausible that regular exercise may serve as a cardioprotective intervention for post-surgery aortic dissection patients, improving clinical outcomes by attenuating hemodynamic responses at rest and during any given level of exercise [13,14], as suggested by animal models of Marfan syndrome [15]. 

Even though there are potential benefits of exercise in this population, few data are available regarding the specific types and intensities of exercise [13]. Physical training of moderate intensity seems feasible and beneficial in post-surgery ATAAD patients [10]. The American Heart Association encourages daily moderate activity, corresponding to three-to-five metabolic equivalents (METs), for these patients to improve their global cardiovascular health and BP control [16,17]. 

However, the impact of cardiac rehabilitation (CR) in these patients is still understudied [18]. The Sport Rehabilitation Exercise Group of the French Society of Cardiology recommends CR as a Class IIa indication in patients with thoracic aortic surgery (level of evidence C) [7]. Finally, the European Society of Cardiology guidelines on aortic diseases state that leisure sport activities with low static/low dynamic stress are acceptable, without specifically discussing CR [2]. Despite these general guidelines, there is still insufficient evidence-based data to support the delivery of CR in this specific population.

The purpose of this study was to assess the impact of an exercise protocol, combining segmental muscle strengthening and moderate-intensity continuous cycling training in ATAAD patients during a comprehensive CR program. 

## 2. Materials and Methods

### 2.1. Study Design

This is a retrospective study in a single center. All patients provided written informed consent approved by Corentin Celton Hospital for undergoing CR, and for the use of their anonymized data.

### 2.2. Study Population

We analyzed the cases of all 75 patients with ATAAD who underwent surgical treatment at Public Assistance–Paris Hospitals, and who were referred to the department of CR and secondary prevention of Corentin Celton Hospital from 1 May 2015 to 15 May 2021. The CR program is open to patients who have undergone ATAAD surgery, excluding unstable postoperative conditions (significant aortic valve regurgitation, thrombosis, or malperfusion), contra-indications for exercise according to the national guidelines of CR including pericardial effusion grade >2 [19], and refusal of informed consent. Two patients were excluded because they were not stable enough to perform CR due to post-operative complications. All of the other patients were discharged in good condition after the standard CR program, and underwent baseline and final evaluations. Therefore, the final study population included 73 patients. 

### 2.3. Cardiac Rehabilitation Program

At baseline, all patients underwent a clinical assessment, blood sampling, ECG, chest X-ray, and echocardiography at rest and an incremental symptom-limited exercise stress test (ExT). The comprehensive CR program mainly included exercise training, treatment optimization (e.g., drug dosage and type, nutrition, pacemaker or ventricular assist device adjustment, etc.), patient education (e.g., individual and group-based programs, including workshops about the disease, diet, BP management, self-monitoring, stress management, and lifestyle counselling, etc.), and psychological and social counselling if required.

### 2.4. Study Procedures

#### 2.4.1. Exercise Stress Test

An incremental submaximal symptom-limited ExT on a cyclo-ergometer was performed before the beginning of CR and at discharge to evaluate exercise capacity, and to guide the subsequent exercise training intensity. The ExT protocol was personalized according to the clinical evaluation, aiming for a duration of exercise between 5 and 10 min. Heart rate (HR), ECG, and arterial oxygen saturation (SaO2) were continuously monitored. BP was measured manually (Tango M2, SunTech Medical, TSC, Flaxlanden, France) at 1-min intervals during the ExT, and during the first 3 min of recovery. The exercise capacity was calculated in METs according to the following equation: METs = ((1.74 × [watts × 6.12/kg]) + 3.5)/3.5 [20].

#### 2.4.2. Exercise Training

Training sessions were programmed for at least 1 h of exercise 5 days/week, including moderate-intensity continuous training (MICT) on a bicycle, and segmental muscle strength training for 30 min each. These sessions were tailored to each patient based on their clinical condition and the baseline ExT assessment. They could be performed consecutively or separately (30 min in the morning and 30 min in the afternoon). Exercise training was supervised by a cardiologist and a physical therapist.

During MICT, the HR and single-lead electrocardiogram were monitored in real time using a heart rate belt, and BP was measured every 2 min. The exercise intensity was set based on HR for patients with a sinus rhythm, and on effort perception for patients with atrial fibrillation. The target HR zone was calculated using the modified Karvonen formula (HR target = HR rest + 0.8 (HR max − HR rest)) because the majority of patients were treated with beta-blockers. Every session consisted of 5 min of warm up, 20 min of training, and 5-to-10 min of cool down. The workload for the first session was adjusted at 50% of the maximum charge reached during the ExT. The progression in workload was made by 5-to-10 watts increments depending on the HR and Borg scale in the previous session, and re-evaluated every 2–3 sessions.

Segmental muscle strengthening included stretching and lifting exercises of the upper and lower limbs with an energy expenditure adapted to the clinical condition of each patient. According to the results of ExT, patients were classified into different segmental muscle strength training levels. The patient’s perception of effort was assessed during exercise. BP was measured before and after exercise training.

### 2.5. Statistical Analysis

Results are expressed as mean ± SD. *p* values < 0.05 are considered significant. The statistical analysis was done using a repeated-measure ANOVA, with modified paired t-tests. Differences before and after CR were analyzed using a two-way ANOVA. The relationships among variables were investigated using the Pearson correlation coefficient (r).

## 3. Results

Table 1 summarizes the baseline characteristics of the 73 patients included in this study. Patients were mainly male (*n* = 57, 78.1%), middle-aged-to-elderly (62.2 ± 12.7 years old, minimum 32, and maximum 87 years old), and with normal left ventricle ejection fractions (66.6 ± 8.2%). The average BMI was in the normal range (23.9 ± 4.4 kg/m^2^). More than half of the patients had hypertension (54.8%) and smoking history (28.7%) before surgery. One-third of the patients had a sedentary lifestyle (23.3%). Surgical procedures included aortic repair by a prosthetic tube (98.6%) with a combined valve replacement in 26%, aortic valve plastic (37%), and one allograft. The majority of patients (93.1%) maintained a sinus rhythm 3 weeks after surgery. As far as medical treatment is concerned, the majority of patients received beta-blockers (90.4%), more than half (63%) were on angiotensin-converting enzyme inhibitors or angiotensin receptor blockers, and 40.3% were on calcium antagonists.

Table 2 shows the results of the ExT before and after CR, and the parameters measured during cycling training. Patients performed 14.5 ± 4.7 sessions of exercise on a bicycle, and 11.8 ± 4.3 sessions of segmental muscle strengthening. During this period, the intensity of the MICT increased significantly from 27.1 ± 16.1 to 46.8 ± 21.2 watts, a net gain of 19.6 ± 10.2. This is equivalent to an increase from 2 to 3 METs. Maximum SBP/DBP during exercise training were 143 ± 14/88 ± 14 mmHg, which is less than the upper SBP limit of 160 mm Hg. BP treatment management during the CR period was required. Therefore, at the end of the exercise-based CR program, exercise capacities were significantly improved. No significant adverse events were recorded.

Table 3 shows that the gain of exercise capacity after the exercise training program was positively related to the gain of workload during the exercise training (*r* = 0.55, *p* < 0.05) and the gain of HR reserve (*r* = 0.6, *p* < 0.05), but negatively related with renal function (*r* = −0.40, *p* < 0.05) and resting DBP (*r* = −0.46, *p* < 0.05).

## 4. Discussion

Our study confirms that an early and moderate intensity post-operative exercise-based CR is feasible and safe under cardiologic supervision.

The benefits of CR are well established for patients with heart disease. Exercise training increases the maximal (VO_2_peak) and sub-maximal (endurance) capacities; improves endothelial function and myocardial flow reserve; and reduces smoking deleterious effects, body weight, blood lipids, and resting BP [21]. However, the management of patients surviving aortic dissection is mainly focused on BP control through intensive anti-hypertensive medication. Strict BP control is believed to be the single most important factor in the postoperative treatment [11,22], and tight BP control is of pivotal importance for patients’ prognosis [23]. This is why many clinicians are worried about the possible risks associated with the exercise-induced increase in BP during CR in post-surgery ATAAD patients.

In our study population, arterial hypertension (54.8%) and smoking (28.7%) were the most common cardiovascular risk factors, followed by sedentary lifestyle (23.3%). Our patients enter the in-hospital CR program on average 26.2 days after emergency surgery. There are no guidelines and only limited literature regarding the recovery period before entry into a CR program. However, this postoperative interval for entering CR was relatively short compared to the interval reported in Fuglsang’s study [7].

Studies [24,25] on abdominal aortic aneurysms (AAA) suggested that the protective effect of regular exercise on AAA expansion may be due to the suppressive effect on resting BP following an elevation of exercise BP. The exercise-induced chronic lowering of BP effect (post-exercise lowering of BP) following exercise training is multifactorial [26]. In the present study, the mean resting SBP/DBP before CR was 121.6 ± 15.5/70.8 ± 10.5 mm Hg. We did not compare the changes in BP before and after CR because antihypertensive argents, as a confounding factor, make it difficult to evaluate the effect of exercise on BP. Furthermore, an average of 4 weeks of in-hospital exercise rehabilitation, including 14.5 ± 4.7 sessions of bicycle and 11.8 ± 4.3 sessions of segmental dynamic resistance exercises may not be sufficient to observe the benefits of exercise on resting BP, especially if the exercise training is performed at a very low intensity. Nevertheless, based on the literature, it seems that exercise can decrease SBP in this patient population. For example, an international registry of acute aortic dissection data reported by Chaddha et al. [9] showed that 36 months of aerobic exercise training (≥2 times/week) significantly reduced resting SBP. Similarly, Nakayama et al. [25] reported that, after 5 months of CR in postoperative AAA patients, resting SBP decreased from 123 to 115 mm Hg. Interestingly, they also found that the increase in SBP during exercise after CR was an independent risk factor for the acceleration of AAA expansion. In the present study, we did not look at the change in SBP during exercise. This is a parameter that will need to be considered in future studies to determine what constitutes a safe exercise BP threshold for CR patients after ATAAD surgery.

Presently, there are only a few studies addressing the safety and outcomes of CR in ATAAD patients. Corone et al. [27] studied 33 patients who underwent moderate-intensity physical training after surgery for ATAAD. They demonstrated that this type of intervention was feasible and beneficial, with no additional dilation on untreated segments of the descending aorta. Similarly, Chaddha et al. [28] recommended mild-to-moderate aerobic exercise, at 3 to 5 METs, for at least 30 min on most days of the week for patients after ATAAD surgery. Their goal was to achieve a reduction in the resting BP, and to improve global cardiovascular health. Data from the International Registry of Acute Aortic Dissection showed that SBP 36 months after discharge for post-dissection patients engaging in two or more sessions of aerobic activity per week was significantly lower compared with those who did not exercise as much [29]. More recently, Fuglsang et al. [7] did a retrospective study on 29 patients who underwent surgery for ATAAD: 19 of them undertook an exercise-based CR program, and 10 did not. They demonstrated that exercise decreased resting BP in both normotensive and hypertensive patients. CR also increased the peak oxygen uptake and maximal workload in all patients. Finally, patients in the CR group displayed a higher health-related quality-of-life [7].

In our study, we observed that the gain of workload during ExT was positively associated with the maximum BP during cycling (Table 3). Similarly, the gain of workload during cycling training was positively correlated to the maximal workload of ExT. It’s interesting to note that in this study the average maximum SBP/DBP during cycling training was 143 ± 14/88 ± 14 mm Hg. This is well below the currently accepted threshold of SBP < 160 mm Hg. This may indicate that the intensity of the training was too low. However, the volume of training, assessed by the overall energy expenditure in kilocalories, can be modulated by adjusting three parameters: exercise intensity, duration, and frequency. Therefore, it may be possible to increase the volume of training to gain more benefits without increasing intensity (i.e., increasing BP) by modulating the duration and frequency of the exercise prescription.

In this study, we also added strength training to the MICT cycling regimen. Again, there are very few studies on the safety and efficacy of strength training in CR after ATAAD surgery. Corone et al. [27] indicated that 16 patients carried out isolated segmental dynamic resistance exercises in addition to aerobic training, and Fuglsang [7] also presented a muscle strength training and stretching program for CR. The impact of these exercise regimens is difficult to quantify and compare with standard resistance training or other patient populations, considering that surgical intervention changes the anatomy and risk factors of ATAAD patients. Our study population performed the segmental dynamic resistance exercises as a supplement to aerobic training while maintaining a BP < 160/100 mm Hg. The exercises were well executed, and no adverse events were observed.

It is also possible that an SBP of <160 mm Hg may not be the optimal threshold for exercise BP in the ATAAD population. In the study of Corone et al. [27], 25% of postoperative ATAAD patients had an exercise SBP > 170 mm Hg. In our study, during the first ExT, 10 patients had an SBP ≥ 180 mm Hg, and 5 patients (4 men and 1 woman) had an exaggerated SBP response, defined as a peak SBP of >210 mm Hg for men and >190 mm Hg for women. During the ExT performed before discharge, seven patients had an SBP ≥ 180 mm Hg, and one woman had an exaggerated SBP response to exercise testing. During exercise training, six patients had an SBP ≥ 160 mm Hg, but none had an SBP ≥180 mm Hg. It is, however, possible that the actual SBP in the study by Corone et al. [27] or in our study exceeded these thresholds between BP measurements. In the limited studies on exercise in aortic dissection patients, the exercise-induced rise in BP did not result in reported adverse events.

The quality-of-life of these patients should also be an important part of our evaluation. Chaddha et al. [9] indicated that 32% of patients after ATAAD reported new-onset depression, and 32% reported new-onset anxiety. Exercise training has been shown to reduce depression and anxiety, thereby improving quality-of-life [21].

The findings of our study are similar to the rare publications on the subject. We confirmed that exercise training improved the intensity of exercise and the maximal workload of these patients without any adverse event. Nevertheless, some adjustments to the general guidelines may be required to ensure patient safety. For example, the American Heart Association Scientific Statement [30] on standards for exercise testing indicates that the BP criteria to terminate exercise testing is >250 mm Hg/115 mm Hg, but a more conservative threshold, defined as a peak SBP of >210 mm Hg for men and >190 mm Hg for women [31,32], is preferable for post-operative ATAAD patients. For most aortic aneurysm patients, aerobic training should be performed at an intensity of SBP < 180 mm Hg, whereas a limit of <160 mm Hg is advised for patients at high risk of dissection or rupture (e.g., women, and people with larger aneurysms) [1]. In all cases, we must emphasize the importance of close BP monitoring during the exercise.

CR provides a good platform to ensure the implementation of individualized exercise training programs through patient-centered holistic care, health education, and optimized medication treatment [33,34].

### Study Limitations and Future Prospects

It’s important to note that the main limitation of this study is its retrospective nature.

Due to the small sample size, we could not control for individual differences (e.g., medical history, comorbidities and complications, past physical condition, lifestyle), surgical differences (e.g., underwent valve replacement or not), and differences in the delay between surgery and CR. After expanding the sample size, these possible confounders should be controlled by stratified analysis and multivariable risk adjustment.

Also, training intensity was determines using training heart rates calculated with the Karvonen formula. The exercise intensity progression was based on perceived exertion (Borg) evaluation, as well as heart rate and BP response during cyclo-ergometer. Cardiopulmonary exercise testing may be more appropriate to quantify the cardiovascular response, especially the ventilatory threshold and exercise capacity. This could provide more information in this type of patients, helping optimize their exercise training protocol.

Some important parameters regarding BP and the intensity of segmental strength training were not quantified during this study. Further monitoring during this type of training may be necessary (e.g., monitor upper limb BP during training of lower limbs, and vice-versa).

In our research population, 28.7% of patients with ATAAD had smoking history. It’s well known that smoking can be a confounder when it comes to VO_2_max measurements because VO_2_max is negatively associated with smoking [35,36,37,38,39]. The VO_2_max of individuals who quit smoking is higher than those who continue to smoke [40]. However, no studies have reported on the duration of smoking cessation necessary to improve VO_2_max. The length of stay for phase II CR in our center was 30.4 ± 11.6 days. We are therefore not able to discuss the impact of smoking reduction and smoking cessation on VO_2_max in post-surgery ATAAD patients before and after CR.

We also lack at this time long-term follow-up results after CR intervention. There are also additional follow-up metrics that could be used during and after CR to improve patient management. For example, we could add a 24-h ambulatory BP monitoring during daily activities, a measure of the maximum rate of left ventricular pressure rise (dp/dt max), as well as a Global Physical Activity Questionnaire (GPAQ) and Quality-of-life (QoL) questionnaires during different periods of CR. Imaging follow-up could also be beneficial, if available.

## 5. Conclusions

This study shows that early postoperative CR is feasible and safe in patients with ATAAD. The effect of CR is remarkable, but it requires constant monitoring during training to ensure that BP does not rise above a target of 160/100 mmHg. These results encourage us to further observe the long-term impact of CR with different exercise types in patients with aortic dissection.

## Figures and Tables

**Table 1 jcm-11-02107-t001:** Baseline characteristics.

	Before CR (*n* = 73)
Age (years)	62.2 ± 12.7
Gender (Male)	57 (78.1%)
Height (cm)	172.9 ± 10.1
Body weight (kg)	71.7 ± 17.8
BMI (kg/m^2^)	23.9± 4.4
SBP (mm Hg)	121.6 ± 15.5
DBP (mm Hg)	70.8 ± 10.5
Left ventricular ejection fraction (%)	66.6 ± 8.2
Small pericardial effusion graded by echocardiography	10 (13.7%)
Hemoglobin (g/dL)	10.6 ± 1.3
Serum creatinine (µmol/L)	84.4 ± 49.1
C-reactive protein (mg/L)	60.9 ± 48.5
Hypertension	54.8%
Sinus rhythm	93.1%
Smoking	28.7%
Sedentary lifestyle	23.3%
Surgical interventions	
<48 h delay from first symptom to surgery	89%
Prosthetic Aortic tube	98.6%
Aortic valve replacement	26.0%
Baseline medications	
Anticoagulants	39.7%
VKA	35.6%
DOAC	4.1%
Antiplatelet drugs without VKA	38.3%
ß-blocker	90.4%
ACEI/ARB	63.0%
CCB	40.3%
Diuretics	23.3%
≥3 Antihypertensive drugs	60.3%

Abbreviations: CR: cardiac rehabilitation; BMI: body mass index; SBP: systolic blood pressure; DBP: diastolic blood pressure; DOAC: direct oral anticoagulants; VKA: vitamin K antagonist; ACEI: angiotensin-converting enzyme inhibitor; ARB: angiotensin receptor blocker; CCB: calcium channel blocker (calcium antagonist). Values are mean ± SD.

**Table 2 jcm-11-02107-t002:** Results of training in CR.

	Before CR	After CR
**CR organization**		
Delay between surgery and CR (days)	26.2 ± 17.3	
Length of stay for phase II CR (days)	30.4 ± 11.6	
Sessions of continuous endurance training (times)	14.5 ± 4.7	
Sessions of segmental strength training (times)	11.8 ± 4.3	
**Endurance training**		
Maximal SBP during endurance training (mm Hg)	143 ± 14	
Maximal DBP during endurance training (mm Hg)	88 ± 14	
Workload during endurance training (watts)	27.1 ± 16.1	46.8 ± 21.2 ***
Gain of workload during endurance training (watts)	+19.6 ± 10.2	
**Exercise stress test (ExT)**		
Resting HR during ExT (bpm)	86.1 ± 17.4	76.4 ± 13.3 ***
Maximal HR during ExT (bpm)	105.0 ± 22.8	110.6 ± 17.2
HR reserve	20.2 ± 13.9	33.2 ± 16.8 ***
Resting SBP	115.3 ± 23.2	116.6 ± 24.8
Resting DBP	68.5 ± 13.2	69.4 ± 13.5
Maximal SBP during ExT (mm Hg)	142.1 ± 28.9	169.8 ± 39.0 *
Maximal DBP during ExT (mm Hg)	74.7 ± 19.1	95.7 ± 30.2 **
Maximal workload during ExT (watts)	63.5 ± 28.6	93.0 ± 35.6 ***
Functional capacity (METs)	3.67 ± 0.96	4.94 ± 1.21 ***
Gain of METs	+1.2 ± 0.6	

Abbreviations: CR: cardiac rehabilitation; HR: heart rate; SBP: systolic blood pressure; DBP: diastolic blood pressure; METs: metabolic equivalents. Values are mean ± SD. *: *p* < 0.05, **: *p* < 0.01, ***: *p* < 0.001; before vs. after CR.

**Table 3 jcm-11-02107-t003:** Correlation gain of exercise capacity after endurance training program.

Correlation (r)	Baseline Creatinine	Gain of Workload during Training	Number of Training Session	Baseline Resting DBP	Gain of HR Reserve
Gain of METs after CR	−0.40 *	0.55 *	0.24	−0.46 *	0.6 *

Abbreviations: CR: cardiac rehabilitation; METs: metabolic equivalents; DBP: diastolic blood pressure; HR: heart rate. *: *p* < 0.05.

## Data Availability

The data underlying the results presented in the study are available from Corentin Celton Hospital.

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
