# Peer review of "Evolution of Early Postoperative Cardiac Rehabilitation in Patients with Acute Type A Aortic Dissection"

_jcm, 2022, doi:10.3390/jcm11082107_

Round 1

Reviewer 1 Report

  • The Authors of the manuscript report that early Cardiac Rehabilitation (CR) following surgery for acute type-A aortic dissection is safe and beneficial without adverse events. The Authors should address the following issues:

    • The study is limited by the retrospective evaluation of the enrolled population. The exclusion criteria and non-elegibility for the CR program should be clearly specified
    • Some post-operative echocardiographic findings are required to enhance CR safety evaluation e.g. residual aortic regurgitation, dimension of residual native aortic root, residual false lumen (persistence, dimension and perfusion or thrombosis), malperfusion of epiaortic vessels or distal aorta etc.

Author Response

We thank the reviewers for their constructive comments. We revised our manuscript "Evolution of Early Postoperative Cardiac Rehabilitation in Patients with Acute Type A Aortic Dissection" (jcm-1646379) as best as we could.

Reviewer #1:

The Authors of the manuscript report that early Cardiac Rehabilitation (CR) following surgery for acute type-A aortic dissection is safe and beneficial without adverse events. The Authors should address the following issues:

  1. The study is limited by the retrospective evaluation of the enrolled population. The exclusion criteria and non-elegibility for the CR program should be clearly specified.

Re: We fully agree and thank the reviewer for the suggestion.

We added the exclusion criteria in study population paragraph: The CR program is open to patients who have undergone ATAAD surgery, excluding unstable postoperative condition, contraindication to exercise training and informed consent refused (Page 2, lines 90-94).

We also include the retrospective evaluation as a limit of the study.  

  1. Some post-operative echocardiographic findings are required to enhance CR safety evaluation e.g. residual aortic regurgitation, dimension of residual native aortic root, residual false lumen (persistence, dimension and perfusion or thrombosis), malperfusion of epiaortic vessels or distal aorta etc.

Re: We thank the reviewer for highlighting this weakness. Before and after CR, we routinely performed the echocardiography and recorded: ejection fraction, the residual aortic regurgitation, dimension of residual native aortic root, residual false lumen (persistence, dimension and perfusion or thrombosis) and also pericardial effusion. We don't display the results of all the echocardiographic findings due to space limitations, but we include some of this in the table 1. Additionally, for reviewer information, our surgical department performed a CT scanner screening before CR, to detect complications, malperfusions, thrombosis or significant regurgitation contra-indicating cardiac rehabilitation and so, these patients are not addressed to our CR centre.    

Moreover, echocardiography needs to be rechecked while the state of patient was unstable during CR. The length of stay for phase II CR in our center was 30.4 ± 11.6 days, therefore, we did not further analyze short-term echocardiographic data.

Reviewer 2 Report

Study is interesting and deals with an interesting topic. References are updated and relevant. Manuscript is well written. Conclusions are supported by data. I have no specific concern.

Author Response

We thank the reviewers for their constructive comments. We revised our manuscript "Evolution of Early Postoperative Cardiac Rehabilitation in Patients with Acute Type A Aortic Dissection" (jcm-1646379) as best as we could.

Reviewer #2:

Study is interesting and deals with an interesting topic. References are updated and relevant. Manuscript is well written. Conclusions are supported by data. I have no specific concern.

Re: We thank the reviewer for this encouraging comment.

Reviewer 3 Report

The paper is well written, well presented and is of interest to readers. 
I do not have any major comments. Minor comment/question: 

Almost 30 % of patients were smokers. How effective was the smoking cessation program during CR and could that have had an impact on the results. 

Author Response

We thank the reviewers for their constructive comments. We revised our manuscript "Evolution of Early Postoperative Cardiac Rehabilitation in Patients with Acute Type A Aortic Dissection" (jcm-1646379) as best as we could.

Reviewer #3:

The paper is well written, well presented and is of interest to readers.

I do not have any major comments. Minor comment/question:

  1. Almost 30 % of patients were smokers. How effective was the smoking cessation program during CR and could that have had an impact on the results.

Re: We thank the reviewer for highlighting this weakness and we adapted the STUDY LIMITATIONS AND FUTURE PROSPECTS accordingly (Page 8, lines 297-298). References have also been update (Page 8, lines 315-323). In our research population, 28.7% of patients with ATAAD had smoking history, including ex-smokers and active smokers. After the surgery and during the CR all of patients stop smoking. We agree with smoking can be as a bias for VO2max: cigarette smoking is an independent risk factor for the development of aortic disease, and that VO2max is negatively associated with smoking.(Inbar et al. 1994)(Tchissambou et al. 2004)(Hirsch et al. 1985)(Montoye et al. 1980)(Tobita et al. 1995) The reduction in VO2max was most pronounced for heavy smokers, and the VO2max of individuals who quit smoking is higher than those who continue to smoke.(Suminski et al. 2009) However, until now, no studies have reported on the length of smoking cessation to restore VO2max. The length of stay for phase II CR in our center was 30.4 ± 11.6 days. Therefore, we did not able to discuss the impact of smoking reduction and smoking cessation on VO2max or VO2peak in patients with ATAAD before and after CR.

Round 2

Reviewer 1 Report

The revised version of the manuscript is suitable for publication